

# A fast monitor and real time early warning system for landslides in the Baige landslide damming event, Tibet, China

## Yongbo Wu, Ruiqing Niu*, Zhen Lu

Institute of Geophysics and Geomatics, China University of Geosciences, Wuhan, China

*Correspondence to: Ruiqing Niu,*E-mail address*: rqniu@163.com

**Abstract:** Landslide Early warning systems has been widely used to avoid potential disaster. In this paper, a fast monitoring and real time precursor predication method is proposed to build the early warning systems for specific landslide. The fast

monitoring network in this system uses ad-hoc technology to build rapid site monitoring network consist of Beidou terminals and fracture monitors. The real time precursor predication method based on the KF-FFT-SVM model is conducted to fulfil precursor early warning of in short time. The KF-FFT-SVM model working in this system is established through the analysis of the precursor slide character in deformation data got by the Beidou terminals. The deformation data is considered as the mechanical vibration of specific landslide and the  KF-FFT-SVM model is trained to predicate the occurrence of landslide by

the real time deformation data. This system not only improves the robustness of site monitoring, but also provides an effective early warning method for specific landslide. It is applied in Baige landslide monitoring and results showed that KF-FFT-SVM early warning model can predication the occurrence of landslide with high accuracy. It will make the early warning work for specific landslide more effective and costless, although numerous continuous monitored precursor slide deformation data are needed to trained the model well.

**1 Introduction**

Landslide hazard is the third largest geological hazard in nature after earthquakes and volcanoes. It is also direct affected by human engineering activities. China is one of the countries that suffered most from landslide disasters in the world(Huang, 2007). Espically the Ms 8.0 Wenchuan earthquake of May 12, 2008 in China,which triggered tens of thousands of landslides over a broad area in west China, some of them buried large sections of some towns and dammed the rivers(Dai et al., 2011).

So the research on reduced of property damages and casualties has always been an urgent problem, and early warning systems for landslides have already been operating in many place of the world(Glade and Nadim, 2014; Stähli et al., 2015).

According to the definition of the United Nations International Strategy for Disaster Reduction (UNISDR 2009), an early warning system is defined as "the set of capacities needed to generate and disseminate timely and meaningful warning information to enable individuals, communities and organizations threatened by a hazard to prepare and to act appropriately

and in sufficient time to reduce the possibility of harm or loss." Refer to the above definitions, efficient landslide EWSs should



comprise four main sets of actions(DiBiago and Kjekstad, 2007): (1)Monitoring activities, i.e. data acquisition, transmission and maintenance of the instruments;(2)Analysis and modelling of the phenomenon;(3)Warning, i.e. the dissemination of simple and understandable information to the exposed elements;(4)Effective response of the elements exposed to risk and risk's knowledge.

5     Landslide EWSs can be divided into regional landslide EWSs and single landslide EWSs from scale range. Regional landslide EWSs use the statistic method to determine the threshold by rainfall. These system is applicable for the rainfall induced shallow landslides, and the classification early warning is given according the preset rainfall intensity–duration threshold combined with real-time monitoring of soil moisture(Baum and Godt, 2010; Gariano et al., 2015, 2016; Hong and Adler, 2007; Rosi et al., 2015).

10     For single landslide EWSs, the key to a successful EWS lies in the ability to identify and measure in real time limited but significant indicators, called precursors, which precede a landslide catastrophic failure(Barla and Antolini, 2016).The precursor characters are reflected by the mechanical properties of the landslide which can be measured by instruments. For example, inclinometer for tilt(Dikshit et al., 2018; Lollino et al., 2002), fiber Bragg grating for fissures(Zhu et al., 2017), Ground-Based Synthetic-Aperture Radar, LiDAR, total station, GPS and photogrammetric techniques for deformations(Atzeni et al., 2015; 15 Barla and Antolini, 2016; Jaboyedoff et al., 2012; Malet et al., 2002; Tarchi et al., 2003), geoelectrical monitor for soil moisture(Supper et al., 2014), wire extensometer for rock fracture(Intrieri et al., 2012), etc. These precursor characters are used to make early warning with respective model or integrated models(Thiebes et al., 2014; Yin et al., 2010).

    It is obviously that the warning model should build according to the mechanics and the mechanism of the instability of a landslide. And the predication accuracy of the early warning model rely on the high quality real-time monitoring data. While 20 the implement of monitoring network is durable as geotechnical engineering has to undertake in the hard environment. Futher more, the monitoring network is easy to broken down in the wild, which means the monitor part of landslide EWSs is less robustness(Intrieri et al., 2013).

    In this paper, a fast monitor and real-time early warning system is proposed. In this system, the monitoring part uses the ad-hoc network technology to ensure the robustness of the system. In order to build a monitor network quickly, especially after 25 the first failure of a landslide, the monitoring station only include Beidou terminal and fracture monitor. The early warning part based on Kalman-FFT-SVM method to establish a real time warning model. The system was applied after the Baige landslide first damming event, Tibet, and, successfully, got the critical slip data of the surface moving by Beidou terminal based on China's Beidou Navigation System. Then we use the critical slip data to train the early warning model. The early warning model predicts the following damming event successfully. Practice shows the fast monitor and real-time early warning 30 system has generalization significance.





## 2 Fast monitor system

### 2.1 Traditional monitor system

The structure of traditional landslide monitoring network is shown in Figure 1. All kinds of monitoring sensors are connected with data acquisition terminal (DTU) through Modbus protocol or SDI-12 protocol. DTU, communication module (GPRS/3G/4G) and power supply system constitute a remote measurement unit (RTU). The measuring data is sent to the mobile communication network through the communication module and transmitted to the control center through the public network. In this way, the transmission rate is unstable and the system robustness is poor. Once the communication of a monitoring point breaks down, it means that all sensor data under this monitoring point cannot be returned, resulting in partial paralysis of the monitor system. Therefore, a more flexible and stable networking structure is needed to improve the robustness of the monitor system.

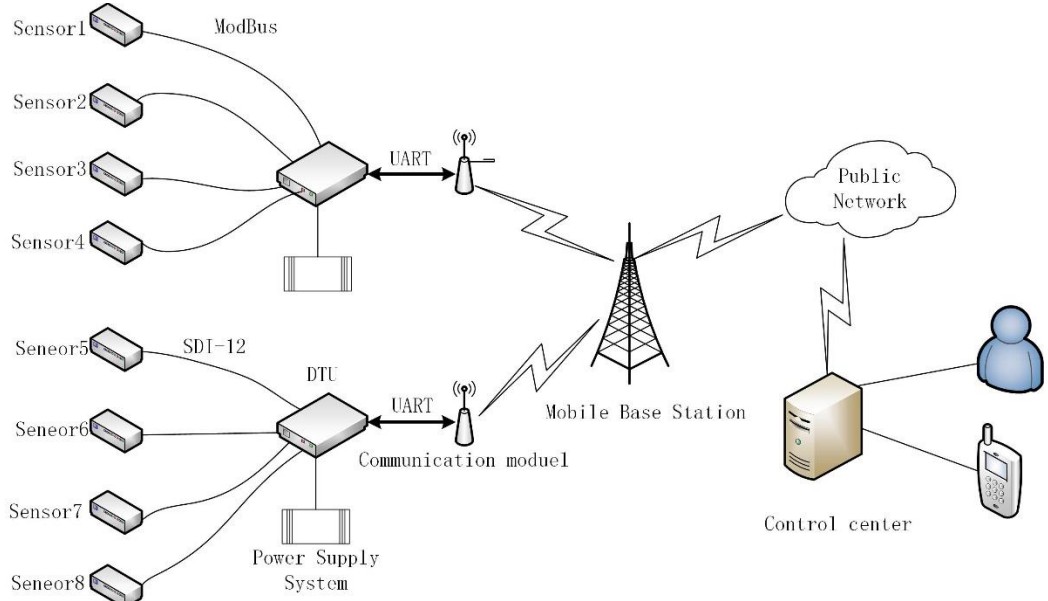

**Figure 1:Traditional landslide monitoring system**

### 2.2 Ad-hoc network  monitoring  system

The adaptive landslide monitoring network is based on ad-hoc network. Ad-hoc network solves the defects of traditional bus and star network, and makes the network more secure, robust, stable and reliable. Figure 2 is a typical structure of adaptive landslide monitoring network. Four stations are listed in figure 2. More stations can be expanded in practical application. Each station is composed of several sensors, data acquisition instrument and ad-hoc router, as shown in Station 1. Each router has the communication module of Beidou/GPRS/3G/4G. At the same time, each router forms a local ad-hoc network through Lora technology. The router can act as an AP (Access Point) node, which is responsible for the access capability of the external

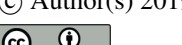


network. Each station also can communicate with the external network through the AP node. At the same time, the routers can also communicate with each other in multi-hops. When a node fails, the network will find other paths to communicate through routing algorithm, which improves the network's robustness .The adaptive landslide monitoring network has three working mode. Normal mode, as is shown in figure 3(a); Communication fault mode, as is shown in figure 3(b). In this mode, parts

5   router is broken down, so the system finds the new routing path to send the data out; Beidou satellite communication mode, as is shown in figure 3(c). This mode means the router is all broken down, so the beidou satellite communication system will be started. The ad-hoc network landslide monitoring system could build on a occurred landslide immediately, especially in the place where have no mobile signal or signal is weak.

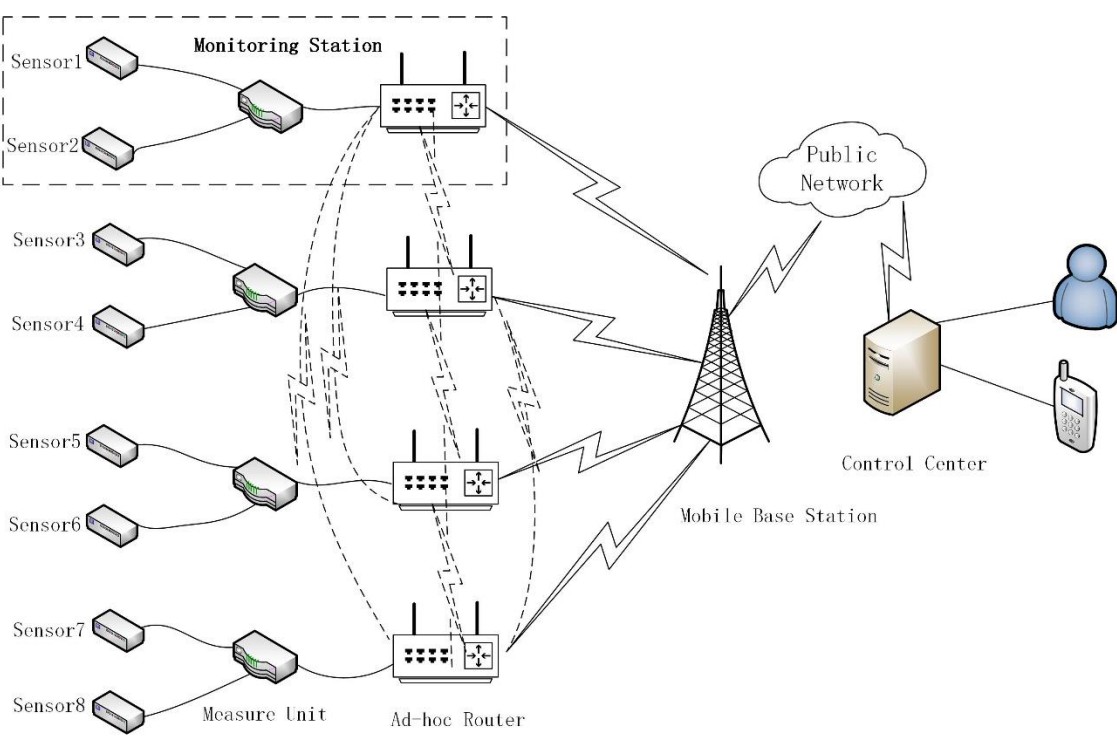

**Figure 2: Ad-hoc network monitoring system**

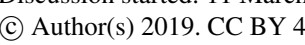


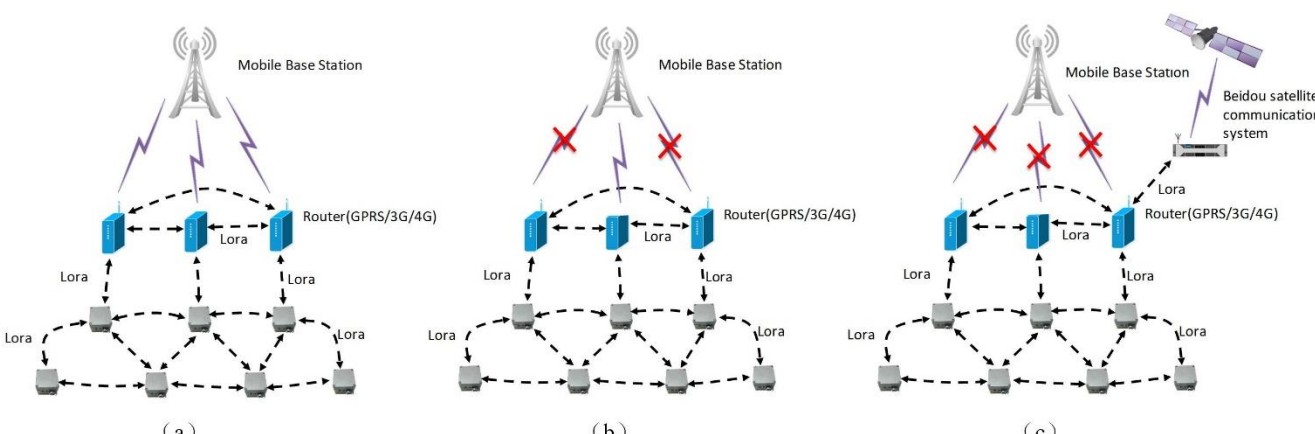

**Figure 3: Adaptive landslide monitoring network working mode. Normal mode (a), Communication fault mode (b), Beidou satellite communication mode (c).**

### 2.3 Application of the system

5       In the early morning of October 11, 2018, a large-scale high-level landslide occurred on the Tibetan Bank of the Jinsha River at the junction of Baige Village, Boro Township, Jiangda County, Tibet Autonomous Region, and Zeba Village, Ronggai Township, Baiyu County, Sichuan Province, blocking the main stream of the Jinsha River and forming a barrier lake. Then, on the late day of November 3, second landslide occurred and blocked the Jinsha River again. The barrier lake formed by the twice landslides caused huge hidden dangers. The location of the landslide is shown in figure 4.



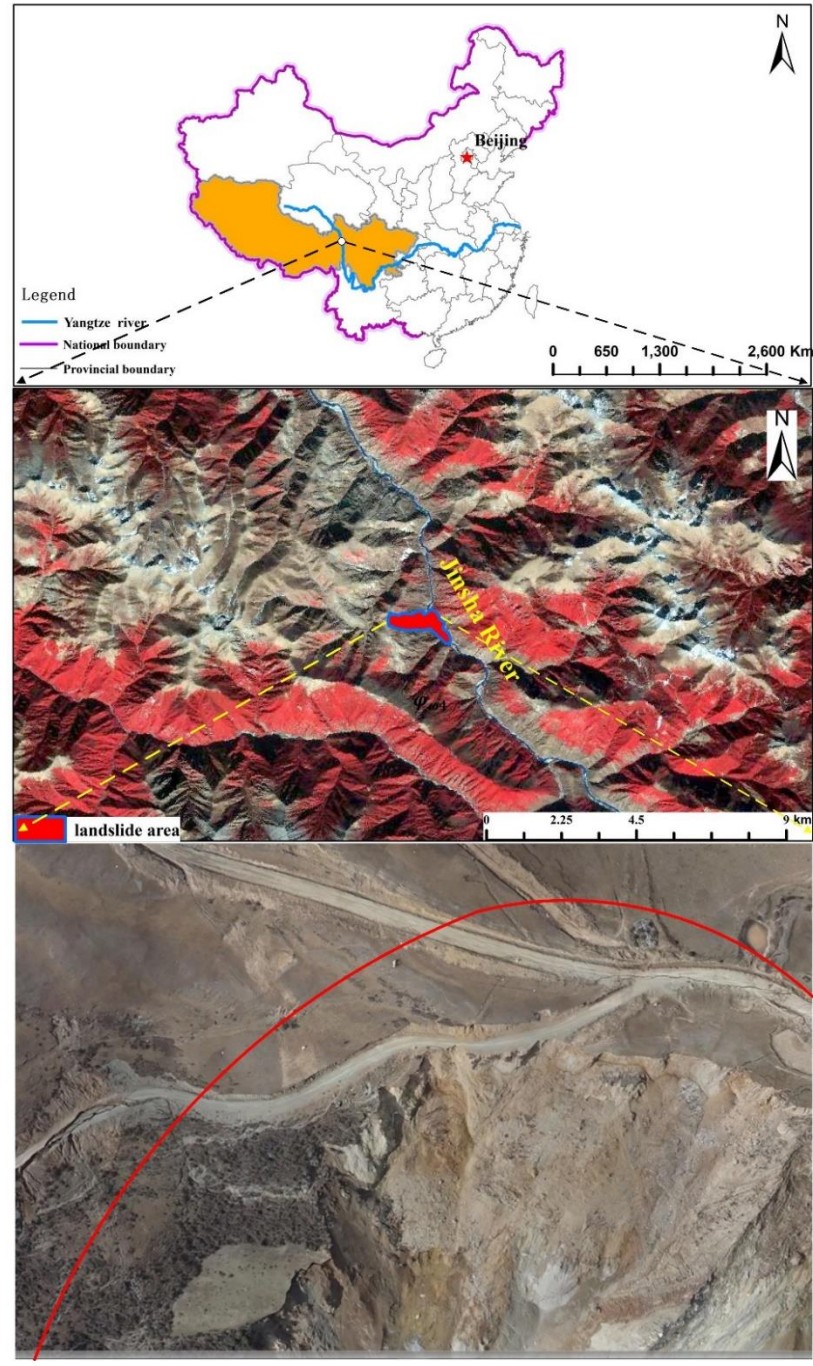

**Figure 4: Location of Baige landslide.**

Baige landslide occurred suddenly, there are no monitor device working there before, meanwhile the monitoring system should be built immediately to ensure the safety of the emergency rescue working for dredging Barrier Lake. As there are no
5    mobile signal there, the fast monitor system is applied there. The fast monitor system contents mainly include surface



displacement monitor and fracture monitor as shown in Figure 5. Figure 5(a) show the beidou receiver on site which is the surface displacement monitoring equipment. Figure 5(b) show the fracture monitor on site which is the fracture monitoring equipment. Both of them use solar panels as an energy supply. The location of the monitor equipment is showed in figure 6, BD1, BD2, BD3 and BD4 is beidou receiver, while Fm1,FM2,F3,and FM4 is fracture monitor.

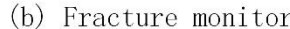

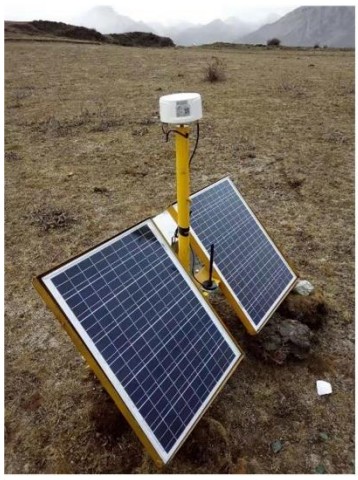 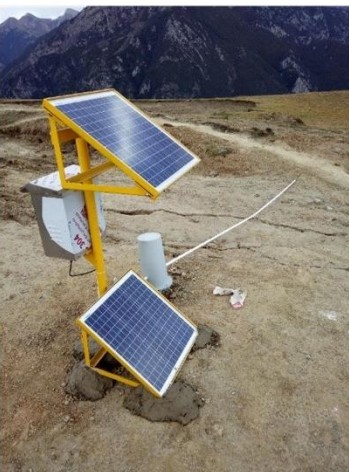

**Figure 5: Monitor device on the landslide.**

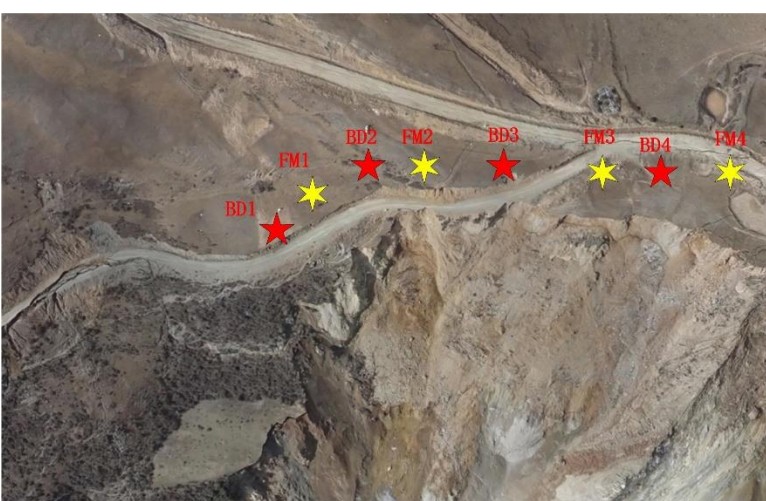

**Figure 6: Equipment locations on the landslide**



## 3 Early warning model

### 3.1 Kalman filtering

Kalman filtering is a linear recursive filtering method based on probability theory and mathematical statistics. It is based on limited data and according to the principle of linear unbiased minimum variance estimation. This method does not need to

store the past observation data. When the new data is generated, the best estimation of the current data can be calculated by using the state transition equation of the signal itself and the recursive formula based on the estimated value of the previous moment and the observed value of the present moment. Kalman filter was put forward by R.E. Kalman in 1960. He introduced the concept of state space into the filtering theory. With the help of the state transition equation of the system, a recursive method was adopted to estimate the new states and observations according to the estimated values at the previous moment and

the observed values at the present moment.

Given a discrete time system, and we have $X_1, X_2, X_3, \cdots, X_k$ as the system state vectors in $kT_s$ , where $X_k \in R^n$ , $T_s$ is the measuring interval.  Define the system control input $U_k$, the incentive noise  $W_k$ . Then, the stochastic difference equation of system state is describe in equation 1.

$$X_k = AX_{k-1} + BU_{k-1} + w_{k-1}. \tag{1}$$

Define the observation variable $Z_k \in R^n$, observation noise $V_k$, we get the observation formula:

$$Z_k = HX_{k-1} + v_k. \tag{2}$$

*A, B, H* is state transition matrix, $w_k, v_k$ are independent normal distribution white noise:

$$w_k \sim N(0, Q) \tag{3}$$

$$v_k \sim N(0, R) \tag{4}$$

In the discrete system state estimating, the formula (1) is used to give the value of $\hat{X}_{j|k}$, which is the best estimating value of  $X_j$ in time $jT_s$ . So there are there situation in the use of formula (1):

a. When $j{=}k$, $\hat{X}_{j|k}$ is the optimum filtering of  $X_k$;

b. When $j{>}k$, $\hat{X}_{j|k}$ is the optimum predicting of  $X_k$;

c. When $j{<}k$, $\hat{X}_{j|k}$ is the optimum smoothing of  $X_k$;

The solve of kalman filtering can describe as time update process and state update process. Time update process:

$$\hat{X}_{k|k-1} = A\hat{X}_{k-1|k-1} + BU_{k-1} \tag{5}$$

$$P_{k|k-1} = AP_{k-1|k-1}A^T + Q \tag{6}$$

Where *P* is error estimating matrix:

$$E_k = X_k - \hat{X}_k \tag{7}$$


$$P_k = E(E_k E_k^T) \tag{8}$$

State update process:

$$K_k = P_{k|k-1}H^T(HP_{k|k-1}H^T + R)^{-1} \tag{9}$$





$$\hat{X}_{k|k} = \hat{X}_{k|k-1} + K_k(Z_k - H\hat{X}_{k|k-1}) \tag{10}$$

$$P_{k|k} = (I - K_k H)P_{k|k-1} \tag{11}$$

## 3.2 Fast fourier transform

Fast Fourier transform(FFT) is a highly efficient algorithm of Discrete Fourier Transform(DFT).Given finite length
sequence x(n), length N and the DFT transform is:

$$X(k) = \sum_{x=0}^{N-1} x(n)W_N^{nk} \tag{12}$$

FFT uses the symmetry, periodicity and reducibility of $W_N^{nk}$ in formula (12), i.e. (2) ~ (4), to decompose a large point DFT into a combination of several small point DFTs. Following is a time-based 2-FFT algorithm.

$$(W_N^{nk})^* = W_N^{-nk} = W_N^{(N-n)k} = W_N^{n(N-k)} \tag{13}$$

$$W_N^{nk} = W_N^{(N+n)k} = W_N^{n(N-k)} \tag{14}$$

$$W_N^{nk} = W_{mN}^{mnk} = W_{N/m}^{nk/m} \tag{15}$$

Given the $N=2^M$, then divide x(n) into 2group. When n is even numbers, let n=2r. When n is odd numbers, let n=2r+1. Let x(2r)=x$_1$(r), X$_1$(k)=DFT[x$_1$(r)], x(2r+1)=x$_2$(r), X$_2$(k)=DFT[x$_2$(r)], where r=0,1,…,N-1. Then formula (12) can be describe as:

$$X(k) = X_1(k) + W_N^k X_2(k) \tag{16}$$

$$X(k + N/2) = X_1(k) + W_N^k X_2(k) \tag{17}$$

It can be calculated that a N-point FFT operation needs NlogN complex multiplication and NlogN complex addition, which greatly improves the operation efficiency of DFT.

## 3.3 Support Vector Machine

Support Vector Machine (SVM) is a statistical Learning Method for Constructing the Optimal Hyperplane based on the principle of structural risk minimization. It maps input vectors into high-dimensional feature space by non-linear
transformation, and then find the optimal classification hyperplane in high-dimensional feature space, which separate the two types of data points as many as possible, and maximum classification interval at the same time. Suppose a training sequence $\{x_i, y_i\}$; i=1,2,…,l; $x_i \in R^n$, $y_i \in \{-1, +1\}$; $l$ is the number of sample, $n$ is the dimension of $x_i$. In the case of linear separability, a classification hyperplane $wx + b = 0$ can be found to separate 2 samples completely. For nonlinearity situation, it should be mapping from the low dimension feature space to a high dimension feature space by a nolinear maping function $\Phi(x)$. Then
the classification hyperplane can be expressed as $w\Phi(x) + b = 0$. Where $w$, b is the variable to be determined. Find the classification hyperplane equivalence maximize $2/\|w\|$. This problem can be solve by Lagrange multiplier method





$$
\begin{cases}
\min \dfrac{\|\boldsymbol{w}\|^2}{2} + C \sum_{i=1}^{l} \xi_i, \\
s.t. \quad y_i(\boldsymbol{w} \cdot x_i + b) \geq 1 - \xi_i, \\
\qquad \xi_i \geq 0, i = 1,2 \dots, l.
\end{cases}
\tag{18}
$$

Where $\xi_i$ is relaxation factor, $C$ is penalty factor. Its duel problem is given by KKT(Karush-Kuhn-Tucher):

$$
\begin{cases}
\max \sum_{i=1}^{l} \alpha_i - \dfrac{1}{2} \sum_{i=1}^{l} \sum_{j=1}^{l} \alpha_i \alpha_j \, y_i y_j \, \boldsymbol{\varphi}(x_i) \cdot \boldsymbol{\varphi}(x_j) \\
s.t. \qquad 0 \leq \alpha_i \leq C, \sum_{i=1}^{l} \alpha_i \, y_i = 0.
\end{cases}
\tag{19}
$$

Solve the problem by SMO algorithm , we can get the classification function:

$$
f(x) = \operatorname{sign} \left\{ \sum_{i=1}^{l} \alpha_i y_i [\boldsymbol{\Phi}(x_i) \cdot \boldsymbol{\Phi}(x_i)] + b \right\}
\tag{20}
$$

Kernel function $K(x_i, x_i) = \boldsymbol{\Phi}(x_i) \cdot \boldsymbol{\Phi}(x_i)$ can be found, which simplified the function operation to inner product of

vectors. The commonly used kernels are linear kernel function, polynomials kernel function, radial basis (Gauss) function

5    and sigmoid kernel function.

### 3.4 Proposed KF-FFT-SVM landslide early warning model

Landslide can be treat a multi-dimensional nonlinear dynamic system influenced by various factors(Eid, 2014).  Many

research predicates the deformation landslide based on these various factors. In this paper, there are not many prior knowledge

about the Baige landslide. The only quantitative data we got is the deformation monitored in infinite time. So in the KF-FFT-

10   SVM landslide early warning model, firstly, we used the infinite deformation data sequence to build the kalman filtering

predication model of the deformation, the formula (21) is the precision evaluation of kalman filtering predication model.

Secondly, we use FFT to analysis the spectrum characteristics of the deformation acceleration nearby the second slide of Baige

landslide on November 3, and find the precursor character of the Baige landslide. Finally, form the acceleration sequence

vector training data and testing date according to the precursor character, and then, label them train the SVM model with the

15   data, the precision of classification result is given as formula (22). The whole process of KF-FFT-SVM landslide early warning

model is showed in figure 7.

$$
\mathrm{RMS} = \frac{1}{M} \sqrt{\sum_{i=1}^{N} (X_i - Z_i)^2}
\tag{21}
$$

$$
Accuracy = \frac{Right \ classification \ numbers}{whole \ samples}
\tag{22}
$$





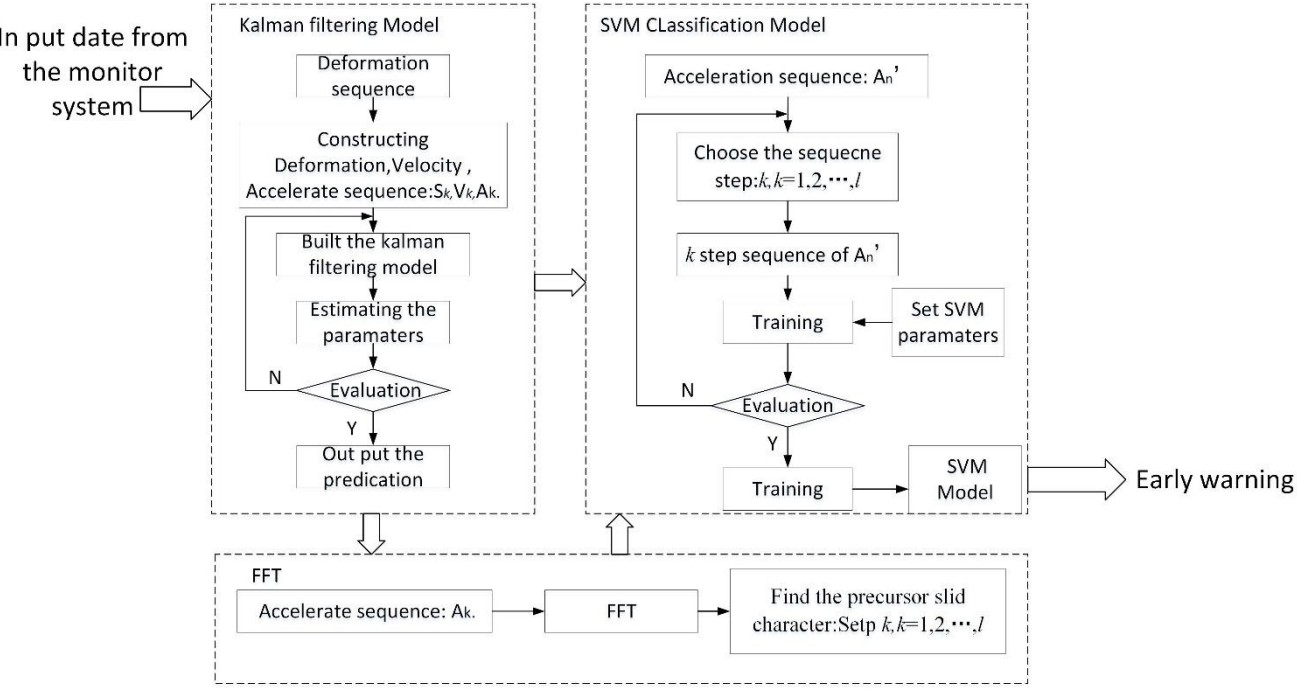

**Figure 7: Process of KF-FFT-SVM landslide early warning model**

## 4 Application of KF-FFT-SVM model

### 4.1 Data pre-processing

5    The deformation of Baige landslide after its first slide is got by beidou receivers which shows in figure 6. The second

landslide occurred on late November 3, so we choose the date from October 31 to November 6 to train the early warning

model. Figure 8 show the raw data of BD1, BD2, BD3 and BD4. The raw deformation is got in the interval of 10 minutes.

It is know from fig.8 that there is noise in the data and the deformation got in much point in 10 minutes interval is static.

So we make a 30 minute statistics, which is show in figure 9. The statistical data is used to build the kalman frittering

10    model.





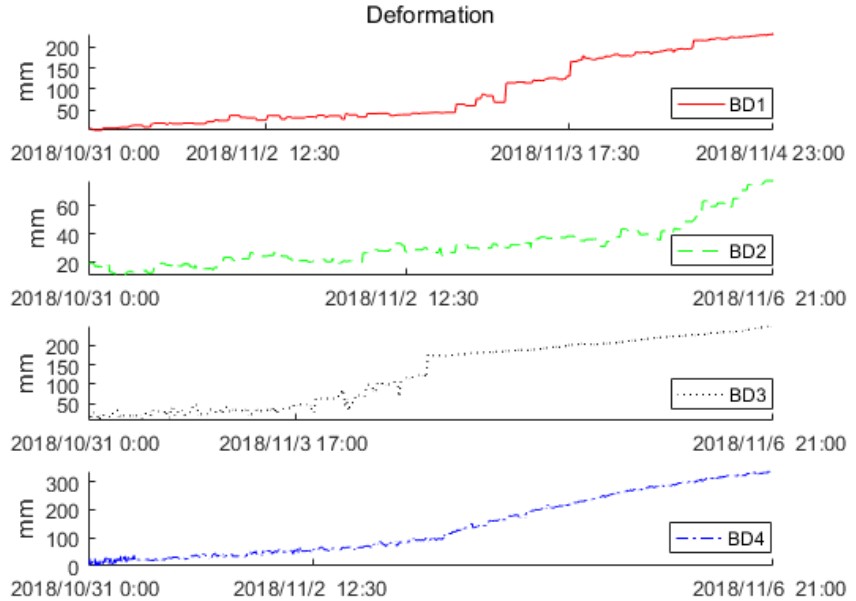

**Figure 8: Raw data of deformation.**

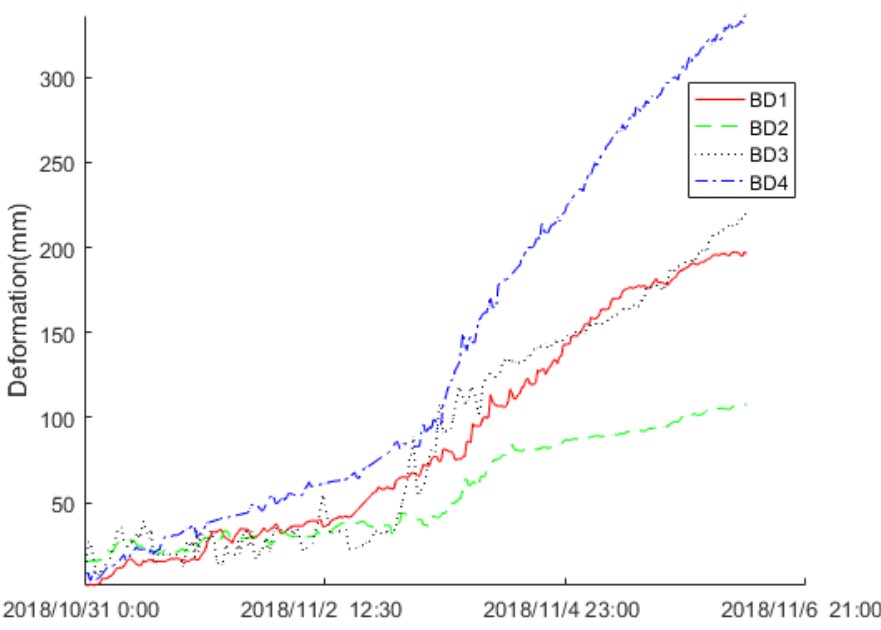

**Figure 9: 30 minutes statistics of the raw deformation data**




## 4.2 KF-FFT-SVM model analysis

### 4.2.1 KF model build

In built landslide deformation KF model, choose deformation, velocity and accelerate as the system state vector, which is $S_k, V_k$ and $A_k$ respectively .The relation between them is:

$$\begin{cases} X_k = [S_k, V_k, A_k]^T, \\ S_k = S_{k-1} + V_{k-1} \cdot T_s + w_{k-1}^1, \\ V_k = V_{k-1} + A_{k-1} \cdot T_s + w_{k-1}^2, \\ A_k = A_{k-1} + w_{k-1}^3. \end{cases} \tag{23}$$

Where $T_s$ is the data acquisition interval; $w_1, w_2, w_3$ is random error. Let $T_s = 1$, then the the stochastic difference equation of system state is:

$$X_k = \begin{bmatrix} 1 & 1 & 0 \\ 0 & 1 & 0 \\ 0 & 0 & 1 \end{bmatrix} X_{k-1} + [w_{k-1}^1, w_{k-1}^2, w_{k-1}^3]^T \tag{24}$$

The observation formula is described as formula (25):

$$Z_k = [1\ 1\ 0]X_{k-1} + v_{k-1} \tag{25}$$

Where $v_k$ is random error and we got $A = \begin{bmatrix} 1 & 1 & 0 \\ 0 & 1 & 0 \\ 0 & 0 & 1 \end{bmatrix}$, H= [1,1, 0]. The random error $w_k$ and $v_k$ is unknown, our purpose is use formula (5) ~ (11) to determine them. At the beginning, set a random value of Q and R, then use the date in section 4.1 to find a couple value of Q and R that makes the KF model convergence and with high fitting accuracy. Figure 10 is the fitting result of BD1, BD2, BD3 and BD4. In this KF model $w_k=[5,3,3]^T$, $v_k=3$. The max fitting error is 5.73mm which means the built KF model have a good prediction and filtering result.





**Figure 10: Kalman filtering fitting result**

### 4.2.2 FFT analysis

Choose the acceleration date between November 3 and November 4 to conduct FFT analysis. Figure 11 show the FFT result

5    of acceleration date during the precursor stage. The FFT length N is 64 and the acquisition interval is $T_s$. Let $T_s = 1$, the acquisition frequency can be simplified to 1 Hz. In figure 11, there are two major amplitude peak value nearby 0.2 Hz and 0.9 Hz, which means, in time domain, the precursor slide character period is nearby $5T_s$. So we choose the step sequence $k = 2, 3, 4, 5, 6, 7, 8$ to construct acceleration sequence.





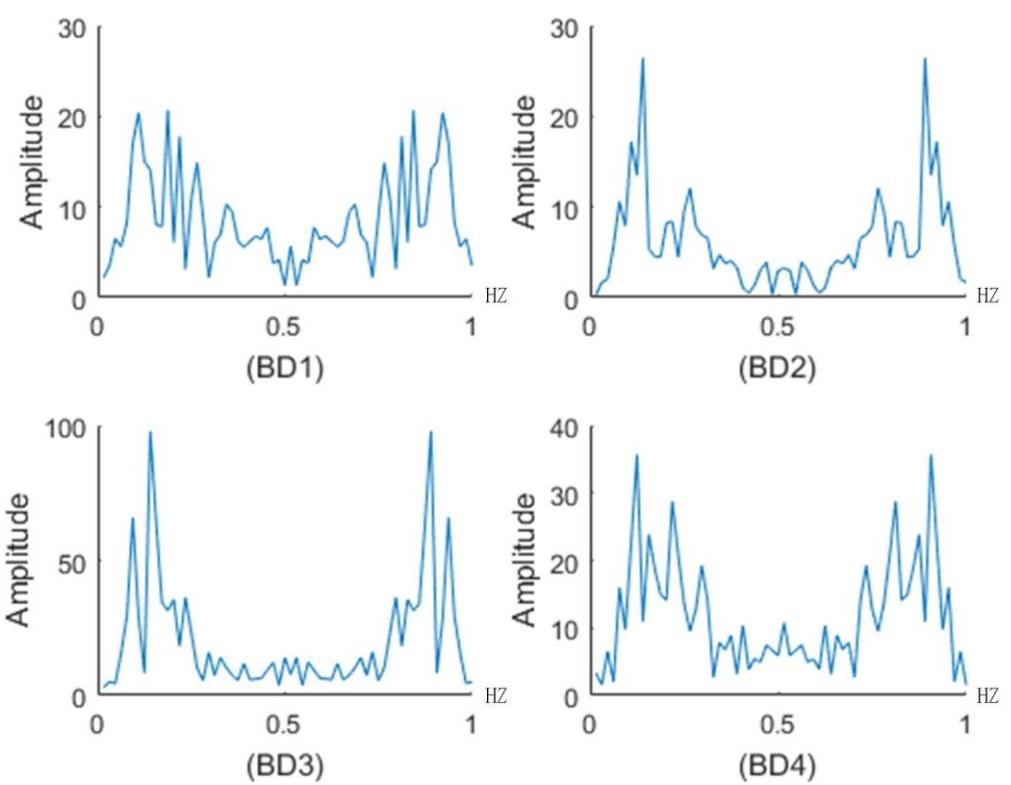

**Figure 11: FFT of precursor slide character**

### 4.2.3 SVM model training

Before SVM model training, we mark the acceleration data of BD1, BD2, BD3 and BD4 manually. The data between
November 3 and November 4 are marked with label "+1" which represents the precursor slide character, others are marked
with label "-1" which represents the non-precursor slide character. Then, use marketed BD1, BD2, BD3 and BD4 data
respectively to construct acceleration sequence $A'_n$,

$$A'_n = [A_n, A_{n+1}, \dots, A_{n+k-1}, label] \tag{26}$$

where $k = 2,3,4,5,6,7,8$, which is given in part 4.2.2; $n = 1,2,\dots,336 - k$; label is marked value of $A_n$.

Then we get the marked set $A'_{n(BD1)}$, $A'_{n(BD2)}$, $A'_{n(BD3)}$, and $A'_{n(BD4)}$. Choose $A'_{n(BD1)}$, $A'_{n(BD2)}$ and $A'_{n(BD3)}$ as training
data, while $A'_{n(BD4)}$ as testing data. For example, when $= 2$, the training data and testing data is showed in formula (27):

$$\begin{cases} & A'_{n(BD1)} = [A_n, A_{n+1}, label]_{(BD1)} \\ training\ data: & A'_{n(BD2)} = [A_n, A_{n+1}, label]_{(BD2)} \\ & A'_{n(BD3)} = [A_n, A_{n+1}, label]_{(BD3)} \\ testing\ \ data: & A'_{n(BD4)} = [A_n, A_{n+1}, label]_{(BD4)} \end{cases} \tag{27}$$





### 4.3 Predicting result

Use RBF function and SMO algorithm to search the best $C$ and γ.The predicting result at different steps sequence is showed in figure 12. Figure 12 also show the 3 steps sequence training data scatters in 3D coordinate and it is obvious to know that they cannot be well separated in 3D space. So we should separate them in high dimension, which means k>3. From figure 12, it is known that when k=6, the highest accuracy 0.915 is got .Meanwhile the best $C$=4, and γ = 1. The result is proved by the FFT analysis which show the best precursor slide character is nearby 0.2Hz, which equal to 6 steps sequence in SVM model.

### 5 Discussion

The fast monitor and real-time early warning system in this paper focus on the monitoring, modelling and waring actions of the efficient landslides EWSs described in part 1. Effective response manners is beyond the discussion of this paper. The use of ad-hoc technology help to layout a redundancy site monitoring network, which improves the robustness of traditional landslide EWSs. The monitor system mentioned here only include Beidou terminals and fracture monitors. So the monitor system can be built in a short time, and it is helpful for the landslide monitoring immediately after the first-time failure of a landslide, because the secondary landslides may occur at any time, which is a great threat to rescuers.

The early warning model based on the KF-FFT-SVM method makes the predication according to the acceleration characters of landslide deformation. It is on the principle that the mechanical vibration of landslide failure can be recorded by the deformation data. Then the precursor acceleration characters is considered as the vibration of landslide failure.  In this study, the Beidou terminals have the ability to measure the deformation of landslide in a short time (10 minutes) and obtain an accuracy of a few millimetres which is given by the manufacture. Raw data of deformation are showed in figure 8, it is obviously that there are random error got by the Beidou terminals as the deformation data is not continuous rise in several period and the deformation is still within the measuring intervals(>10 minutes). That's why the raw data is pre-processed and the 30 minutes statistics of the raw deformation data is, then, given in figure 9. Suppose 30 minutes as one unit time scale, then KF method is used to filter the random error, FFT is used to find the precursor acceleration characters which represent the mechanical vibration frequency of landslide failure. The precursor acceleration characters is finally used to train the SVM classifier and the trained SVM classifier can be set online for real time  landslide early warning.

This system is successfully used in Baige landslide and fulfil the recognition and early warning of secondary landslide. The most important features of the system is that it can quickly build monitoring network and use the deformation data to carry out precursor slide early warning. In this early warning system, we consider the deformation data measured by Beidou terminals as the mechanical vibration of landslide. While, there must be distortion for the transformation from vibration signal to deformation signal, which call for the high sensitivity of Beidou terminals, meanwhile, with high accuracy. The monitor position is also a key to measure the precursor slide characters. In practice it is difficult to put the measuring instrument correctly on the deformation prat which is inner the landslide body. So we locate the Beidou terminals near the surface fracture to make sure measuring the precursor slide characters as possible as it can. Furthermore, the characteristic frequencies got by





the FFT method are different between BD1, BD2 and BD3, then a range frequency is used to generate different length time sequences. SVM model is trained by these sequences respectively and test the accuracy by BD4 to find the best character frequency which can be used to make the early warning for this specific landslide. The training data set in this paper is there, while more Beidou terminals are needed to get more data sets to make this method more effective.

The fast monitor and real-time early warning system is useful for single landslide early warning because it is simplicity, costless, redundancy and robustness. It is especially meaningful for the rescue work of a large scale specific landslide after its first-time failure. Generally, there are limited time left for the rescuing work, so build an effective early warning system in a short time is important. The KF-FFT-SVM model trained by the precursor deformation data of landslide makes the single landslide early warning more effective and it also can be combine with the monitoring of the acoustic emissions from a specific

landslide(Hu et al., 2018).

## 6 Conclusion

In this study, the fast monitor and real time early warning system for landslide is proposed. This system uses ad-hoc technology to facilitate the repaid layout of the site monitoring network, which improves the robustness of traditional landsldes EWSs. Furthermore it builds KF-FFT-SVM early warning model for single landslide through the analysis of the precursor

slide character through the deformation data.

The most important features of the system is that it can quickly layout a monitor network and use the deformation data to carry out precursor slide early warning, this is very useful for the early warning of the specific landslide after its first-time failure. It provides a new idea for monitoring and early warning of single landslide, which not only improve the robustness of the landslide early warning system but also makes landside warning not depend too much on the study of landslide mechanism

characteristics, simplifies landslide monitoring elements. The precursor character extracts from the deformation data is considered as the mechanical vibrations of the landslide failure. Then the real time early warning is conducted according to the precursor slide deformation data.

**Acknowledgements.** This paper was prepared with the help of the Monitoring of Baige Landslide in Jinsha River Project,

supported by Institute of Geospatial Information, China University of Geosciences. It is also partially supported by the by the Geological Survey Project (No. 0431203), the Three Gorges Follow-up Work on Geological Disaster Prevention and Research Project (No. 0001212018CC60010, 0001122012AC50021).





**Figure 12: Prediction result at different steps sequence**



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
