# Peer review of "A fast monitor and real time early warning system for landslides in the Baige landslide damming event, Tibet, China"

_Natural Hazards and Earth System Sciences, 2019_

## Referee Comment (RC1) · Anonymous Referee #1 · 29 Apr 2019

The MS describes a monitoring and warning system for the Baige landslide, which occured twice. Two types of sensors were installed, fracture sensors and GPS sensors, using the Beidou satellite system. The first part of the MS handles the overall concept of the system, whereas the second part describes the models and algorithms used to detect the slide. Comments: There is only a poor connection from the first part to the second one. It remains unclear, why this kind of installation was used and why only one type of sensors was used in the second part. There are no references to other systems taht are on the market, which act in a similar way.

Figure 4: Please indicate the fault scarp and describe why the sensors has been in-

[Figure]

stalled at the specific locations

How are the data monitored (measuring interval versus recording intervall, accuracy of the sensor data

The secon partt oft he MS starts with a desription of methods used. At this time the reader does not know why the different methods were used and what is the benefit of combining these methods instead of just using the Beidou data?

What data are really measured? Horizontal displacement, spatial displacement (x,y,z)..? Figure 8: time axis is wrong, mention the time of failure What does 30 minutes statistics mean (running average?), also in connection with the warning time (Fig. 9). What is the difference in using raw data or averaged data?

Kalman filtering chapter: Please check the indices (seems to be wrong) Chapter 3.4.: what does . . .deformation monitored in infinite time... mean? Fig. 7: Seems to be the key of the MS: Please indicate and describe the steps: What is the input to one model, what is the output, how to integrate the results to the next model step . . ..in order to get a warning message. This is the method and has to be clearly presented. Why you used displacement, velocity and acceleration. These data are of same origin? Figure 10: velocity and acceleration does not have the unit [mm]. What are the results from BD1 – 4? FFT: Why you chose 64 frequency bands? Hz is defined as number per second, but this is not true in that case. The time base may be 10 or 30 minutes???? SVM training: How to differentiate precursor slide character to other data?

Is there a difference in the data of fracture and Beidou sensors? How long does it take to issue a warning with this system (6*10 oder 30 minutes)?

In summary is seems that this system is quite useful, but the MS is structured in an unfavorable way, leaving the reader confused.

---

## Author Comment (AC1) · 17 May 2019

We are appreciated for the referee's comments and their careful reading of our MS. Please find bellow our answers to all items raised.

1. The MS describes a monitoring and warning system for the Baige landslide, which occured twice. Two types of sensors were installed, fracture sensors and GPS sensors, using the Beidou satellite system. The first part of the MS handles the overall concept of the system, whereas the second part describes the models and algorithms used to detect the slide. Comments: There is only a poor connection from the first part to the second one. It remains unclear, why this kind of installation was used and

why only one type of sensors was used in the second part. There are no references to other systems that are on the market, which act in a similar way. Reply: As for the reason about why this kind of installation is used, we are sorry not mention it clearly. Actually, it is described in section 2.3. There are no monitoring device before and the monitoring system should be built immediately to ensure the safety of the emergency rescue working for dredging Barrier Lake which is great threat to the life of millions people living downstream of Jinsha River. The two types of sensors mentioned in this paper take a short time to installation and need not do much geotechnical engineering work comparing with other sensors. Meanwhile, this place has no phone signals, so the Beidou satellite communication system is used to transmit the measure data outside. Ad-hoc network is used to build the on-site monitoring system, which on the one hand, improves the system robust, and one the other hand, needs less Beidou communication terminal unit. That why we choose this system with only two types of sensors and on-site ad-hoc network to undertake the monitor mission. We have look forward to the recently references that concern the Landslide Early Warning Systems(LEWS), there sure have no other similar system on the market, but the technical used in the system is mature and it can be called an integrated innovation. As to the relationship of the first part and the second part, our opinion is that the second part is supplement of first part. The precursor character early warning is always an unsolved problem in the LEWS according references. So we try to build an early warning model that can find the precursor character of specific landslide. The model discussed in this paper is a try to do that only with the deformation information of specific landslide. So the connection between the two parts is that the fast monitor system in the first part provide only the deformation data of specific landslide and the second part describes how to use the deformation data to make precursor character early warning.

2. Figure 4: Please indicate the fault scarp and describe why the sensors has been installed at the specific locations. Reply: Thanks again for the valuable suggestion. The fault scarp has been marked out in figure 4 in red bold line. Also an explanation of why the sensor has been installed at the specific locations is add at the end of

section 2.3 which is: "The sensor is located nearby the back edge fault scarp, neither too far away nor too close. Because too far away from the fault scarp the deformation of landslide got by the sensors is not real, while too close the installation work can be dangerous."

3. How are the data monitored (measuring interval versus recording interval), accuracy of the sensor data. Reply: The measuring interval and recording interval is 10 minutes. The Beidou receiver accuracy given by the manufacture are horizontal direction within 2mm and vertical direction within 3mm. The fracture monitor accuracy given by the manufacture is within 1mm.

4. The second part of the MS starts with a description of methods used. At this time the reader does not know why the different methods were used and what is the benefit of combining these methods instead of just using the Beidou data? Reply: The reason we use these methods is that we want to find the precursor character of specific land-slide implicit in deformation data measured by Beidou receiver.

5. What data are really measured? Horizontal displacement, spatial displacement (x,y,z)..? Figure 8: time axis is wrong, mention the time of failure What does 30 minutes statistics mean (running average?), also in connection with the warning time (Fig. 9). What is the difference in using raw data or averaged data? Reply: In this paper the horizontal deformation data is used. Time axis in figure 8 have been corrected. The 30 minutes statistics means 30-minute interval sampling not just running average and the warning time at the follow discussion is under the basic unit of 30-minute. The raw data is sampled at the interval of 10 minutes and the deformation of specific landslide in 10 minutes may be 0 mm so it cannot reflect the dynamic character of specific landslide. If we use the 30 minutes statistic data the deformation data of specific landslide is a dynamic curve in Fig.9.

6. Kalman filtering chapter: Please check the indices (seems to be wrong) Chapter 3.4.:what does …deformation monitored in infinite time... mean? Fig. 7: Seems to be

the key of the MS: Please indicate and describe the steps: What is the input to one model, what is the output, how to integrate the results to the next model step. . ..in order to get a warning message. This is the method and has to be clearly presented. Why you used displacement, velocity and acceleration. These data are of same origin? Figure10: velocity and acceleration does not have the unit [mm]. What are the results from BD1 − 4? FFT: Why you chose 64 frequency bands? Hz is defined as number per second, but this is not true in that case. The time base may be 10 or 30 minutes???? SVM training: How to differentiate precursor slide character to other data? Reply: We have check the indices in Kalman filtering chapter, there sure have some mistakes. "infinite time" in chapter 3.4 should be "finite time", it is a spelling mistake, we have corrected it. Fig.7 has been redrawn and the input, output of the model is clearly described in chapter 3.4. The displacement, velocity and acceleration is used to build the Kalman filter model, they all derive from the beidou receivers in this paper. The unit of velocity and acceleration should be mm/Ts and mm/(Ts)ˆ2 respectively. Fig.10 is redrawn. The result from BD1 − BD4 is draw in red curve in Fig.10 while the blue curve is the original data. We have chosen 64 numbers of acceleration data between November 3 and November 4,which is the time around the secondary landslides happening, to conduct FFT analysis. So 64 is not the frequency bands, actually it the FFT Length. In this paper, we simplify the sampling interval(30-minite) to 1s, so the biggest frequency band can be treat as 1Hz, we think it is more easy to express the FFT result by this way. In the SVM training, we made the precursor slide character sequences according FFT result, for example if the FFT give a character frequency of 0.2Hz, then we choose k=5 to form training sequence A'n like that A'n=[An,A(n+1),. . .,A(n+4),label]. Principle of the label value in there is that the data between November 3 and November 4 are marked with label "+1" which represents the precursor slide character, others are marked with label "-1" which represents the non-precursor slide character. The we train the SVM until it converges, then we use the SVM to give a predication of An and the result Bn is a sequence of "+1" and "-1"."+1" represents the precursor slide date and "-1" represents the other. That how we differentiate precursor slide character to other

data.

7.Is there a difference in the data of fracture and Beidou sensors? How long does it take to issue a warning with this system (6*10 oder 30 minutes)? Reply: Yes, the fracture sensor measures the width of a fracture, it is a relative displacement. The Beidou sensor measures the absolute displacement and it have a big measuring rang than fracture sensor, that's why in the early waring model we only use the Beidou data. It depends on the interval Ts and "step k" , as to the time take to issue a warning. Generally it takes Ts*k to issue a warning. In this paper, Ts=30 minutes, k=5, and it takes 150 minutes to issue a warning.

Please also note the supplement to this comment:
https://www.nat-hazards-earth-syst-sci-discuss.net/nhess-2019-48/nhess-2019-48-AC1-supplement.pdf
* * *
[Figure]

**Fig. 1.**

[Figure]

**Fig. 2.**

[Figure]

**Fig. 3.**

**Supplement:**

[revised manuscript text omitted]
 finite deformation data sequence $S_k$ got by beidou receivers to build the input of kalman filtering predication model $X_k = (S_k, V_k, A_k)$, where $V_k$ and $A_k$ are the velocity and acceleration of $S_k$ , respectively. After the first step we got the predication and filtering result of $A_n$. Formula (21) is the precision evaluation of kalman filtering predication model. Secondly, we use FFT to analysis the spectrum characteristics of the deformation acceleration sequence $A_n$ , got by step one, nearby the slide time of specific landslide, and find the precursor character of the

15    Baige landslide 'Step $k$ ' which represents the frequency character of $A_n$ nearby the slide time. Finally, use $A_n$ form the acceleration sequence vector training data and testing data according to the precursor character 'Step $k$ 'and label them, then train the SVM model with the data and use the trained SVM model to make predication by a new $A_k$, to find out if the warning is made at that time. The predication result $B_n$ is a vector with the same dimension of $A_k$ and its value is either '0'or '1', '0' represent there is no slide warning while '1' represent the slide warning. The precision of classification result is given as

20    formula (22). The whole process of KF-FFT-SVM landslide early warning model is showed in figure 7.

$$\text{RMS} = \frac{1}{M}\sqrt{\sum_{i=1}^{N}(X_i - Z_i)^2} \qquad (21)$$

$$Accuracy = \frac{Right\ classification\ numbers}{whole\ samples} \qquad (22)$$

[Figure]

**Figure 7: Process of KF-FFT-SVM landslide early warning model**

**4 Application of KF-FFT-SVM model**

**4.1 Data pre-processing**

The deformation of Baige landslide after its first slide is got by beidou receivers which shows in figure 6. The second landslide occurred on late November 3, so we choose the date from October 31 to November 6 to train the early warning model. Figure 8 show the raw data of BD1, BD2, BD3 and BD4(In this paper the horizontal deformation data is used). The raw deformation is got in the interval of 10 minutes. It is know from fig.8 that there is noise in the data and the deformation got in much point in 10 minutes interval is static. So we make a 30 minute statistics, which means a 30-minute interval sampling. The 30 minute statistics result is show in figure 9. The statistical data is used to build the kalman frittering model.

[Figure]

**Figure 8: Raw data of deformation.**

[Figure]

**Figure 9: 30 minutes statistics of the raw deformation data**

**4.2 KF-FFT-SVM model analysis**

**4.2.1 KF model build**

In built landslide deformation KF model, choose deformation, velocity and accelerate as the system state vector, which is $S_k, V_k$ and $A_k$ respectively. The relation between them is:

$$\begin{cases} X_k = [S_k, V_k, A_k]^T, \\ S_k = S_{k-1} + V_{k-1} \cdot T_s + w_{k-1}^1, \\ V_k = V_{k-1} + A_{k-1} \cdot T_s + w_{k-1}^2, \\ A_k = A_{k-1} + w_{k-1}^3. \end{cases} \tag{23}$$

Where $T_s$ is the data acquisition interval; $w_k^1, w_k^2, w_k^3$ is random error. Let $T_s = 1$, then the stochastic difference equation of system state is:

$$X_k = \begin{bmatrix} 1 & 1 & 0 \\ 0 & 1 & 0 \\ 0 & 0 & 1 \end{bmatrix} X_{k-1} + [w_{k-1}^1, w_{k-1}^2, w_{k-1}^3]^T \tag{24}$$

The observation formula is described as formula (25):

$$Z_k = [1\ 1\ 0]X_{k-1} + v_{k-1} \tag{25}$$

Where $v_k$ is random error and we got $A = \begin{bmatrix} 1 & 1 & 0 \\ 0 & 1 & 0 \\ 0 & 0 & 1 \end{bmatrix}$, H= [1,1, 0]. The random error $w_k$ and $v_k$ is unknown, our purpose is use formula (5) ~ (11) to determine them. At the beginning, set a random value of Q and R, then use the date in section 4.1 to find a couple value of Q and R that makes the KF model convergence and with high fitting accuracy. Figure 10 is the fitting result of BD1, BD2, BD3 and BD4, see the red curve. In this KF model $W_k=[5,3,3]^T$, $V_k=3$. The max fitting error is 5.73mm which means the built KF model have a good prediction and filtering result.

[Figure]

**Figure 10: Kalman filtering fitting result**

**4.2.2 FFT analysis**

Choose 64 numbers of acceleration data between November 3 and November 4,which is the time around the secondary landslides happening, 
[revised manuscript text omitted]

---

## Referee Comment (RC2) · Anonymous Referee #2 · 3 Sep 2019

SUMMARY: The manuscript describes the technical development and implementation of a monitoring system with data analyses based on Kalman filtering, fast fourier transformation and a support vector machine based on displacement data. The study area is a landslides in the Sichuan/Tibet, China which also caused a landslide dam.

GENERAL COMMENTS: There are many typos and language errors that need revision. Please consider using a language editing service or ask a native speaker for assistance. I think the review of existing landslide early warning systems should be more extensive. I provided some sources. I find that monitoring, forecasting, nowcasting and early warning systems should be distinguished more clearly, e.g. by applying

the UNISDR/UNDRR classification that you cited. The description of the landslide, its geomorphology, trigger, the aftermath (dam and lake) etc is very brief and not appropriate to understand the conditions in the field. The focus clearly lies on the technical development and aspects of warning are not covered. Thus, I recommend to not describe this as an early warning system. The presentation of the research could be made more clear.

SPECIFIC COMMENTS: Title: The title is not in proper English. It should be "monitoring" system, not "monitor". It is also unclear whether this is a local scale system (because you name Baige landslide event) or a regional system (because you use landslides in plural).

Abstract: P1L8: Please avoid capitalisation where not appropriate. "Early", the second word, should not be with a capital "E". P1L8: I would also argue, that an EWS (early warning system) does not really avoid a disaster as it does not stop the landslide from happening. P1L9: for A specific landslide OR for specific landslideS. P1L10: people not familiar with China have no idea what Beidou or a Beidou terminal is. Please rephrase. P1L11: The real time precursor predication method IS based

There are obviously substantial language issues that need to be resolved. Please revise. No further comments on typops and language are provided from my side in the remaining review.

P1L11: there is no explanation what a KF-FTT-SVM model is. What does the abbrevation stand for? P1L13: rather use landslide instead of slide.

Introduction: P1L21: Provide a citation for the claim that landslides are the third largest (?) geological hazard. P1L22: Add a space before the bracket. There are also several instances of missing spaces in the remaining document. P1L26ff: additional information on landslide EWS could be added, consider adding information from

Overview on landslide EWS and review of existing systems: Thiebes, Benni, and

Thomas Glade. "Landslide Early Warning Systems – Fundamental Concepts and Innovative Applications." In Landslides and Engineered Slopes. Experience, Theory and Practice, edited by S Aversa, L Cascini, L Picarelli, and C Scavia, 1903–1911. Naples, Italy: CRC Press, 2016. https://doi.org/10.1201/b21520-238. Thiebes, Benni. Landslide Analysis and Early Warning Systems: Local and Regional Case Study in the Swabian Alb, Germany. Springer Theses Series. Springer, 2012. Case studies: Thiebes, Benni, Rainer Bell, Thomas Glade, Stefan Jäger, Julia Mayer, Malcolm Anderson, and Liz Holcombe. "Integration of a Limit-Equilibrium Model into a Landslide Early Warning System." Landslides 11, no. 5 (June 14, 2013): 859–75. https://doi.org/10.1007/s10346-013-0416-2. Calvello, Michele, Ricardo Neiva d'Orsi, Luca Piciullo, Nelson Paes, Marcelo Magalhaes, and Willy Alvarenga Lacerda. "The Rio de Janeiro Early Warning System for Rainfall-Induced Landslides: Analysis of Performance for the Years 2010–2013." International Journal of Disaster Risk Reduction, October 2014. https://doi.org/10.1016/j.ijdrr.2014.10.005. Michoud, C., S. Bazin, L. H. Blikra, M.-H. Derron, and M. Jaboyedoff. "Overview of Existing Landslide Early-Warning Systems in Operation." In EGU General Assembly Conference Abstracts, 14:2919, 2012. http://adsabs.harvard.edu/abs/2012EGUGA..14.2919M. Piciullo, Luca, Michele Calvello, and JoséÂăMauricio Cepeda. "Territorial Early Warning Systems for Rainfall-Induced Landslides." Earth-Science Reviews 179 (April 2018): 228–47. https://doi.org/10.1016/j.earscirev.2018.02.013. Piciullo, Luca, Mads-Peter Dahl, Graziella Devoli, Hervé Colleuille, and Michele Calvello. "Performance Evaluation of the National Norwegian Early Warning System for Weather Induced Landslides." Natural Hazards and Earth System Sciences Discussions, January 16, 2017, 1–28. https://doi.org/10.5194/nhess-2017-24. Rossi, Mauro, Silvia Peruccacci, M. T. Brunetti, I Marchesini, S. Luciani, Francesca Ardizzone, V Balducci, et al. "SANF: National Warning System for Rainfall-Induced Landslides in Italy." In Proceedings of the 11th International & 2nd North American Symposium on Landslides, edited by E Eberhardt, Corey R. Froese, A. Keith Turner, and S. Leroueil, 2:1895–99. London: Taylor & Francis, 2012. Segoni, S., A. Battistini, G. Rossi, A. Rosi, D. Lagomarsino, F. Catani, S. Moretti, and N. Casagli. "Technical Note: An Operational Landslide Early Warning System at Regional Scale Based on Space–Time-Variable Rainfall Thresholds." Natural Hazards and Earth System Science 15, no. 4 (April 16, 2015): 853–61. https://doi.org/10.5194/nhess-15-853-2015. Kirschbaum, Dalia Bach, Robert Adler, Yang Hong, Sujay Kumar, Christa Peters-Lidard, and Arthur Lerner-Lam. "Advances in Landslide Nowcasting: Evaluation of a Global and Regional Modeling Approach." Environmental Earth Sciences 66, no. 6 (July 2012): 1683–96. https://doi.org/10.1007/s12665-011-0990-3.

P2L10: you presented the UNISDR/UNDRR classification of EWS; however, the systems you mention in this line and the following are not necessarily real EWS that include works on all 4 fields of action. They are mostly monitoring and/or forecasting systems. Please check carefully whether they are really fully fledged EWS, e.g. by checking to which extent they are really in operation (most are not but are monitoring systems with some aspects of predictions). P3 Figure 1: Typos in figure. Seneor should be sensor. P5L8: can you provide more information on the characteristics of the landslide, the dam and the lake? What are the geological and geomorphological conditions? Why was the landslide triggered? Have there been any movements before? P11L5ff: the language is not appropriate. Please rephrase this section.

Discussion: The discussion does not really reflect on the limitations and uncertainties of the study but rather summarises the research. This should be revised.

---

## Author Comment (AC2) · 13 Oct 2019

We are appreciated for the referee's comments and their careful reading of our MS. Please find bellow our answers to all items raised. 1. Title: The title is not in proper English. It should be "monitoring" system, not "monitor". It is also unclear whether this is a local scale system (because you name Baige landslide event) or a regional system (because you use landslides in plural). Reply: Thanks again for the valuable suggestion. The Title has been changed as "A fast monitoring and real time early warning system for landslide in the Baige landslide damming event, Tibet, China". The system mentioned in this manuscript is a local system.

2. Abstract: P1L8: Please avoid capitalization where not appropriate. "Early", the second word, should not be with a capital "E". P1L8: I would also argue, that an EWS (early warning system) does not really avoid a disaster as it does not stop the landslide from happening. P1L9: for A specific landslide OR for specific landslides. P1L10: people not familiar with China have no idea what Beidou or a Beidou terminal is. Please rephrase. P1L11: The real time precursor predication method IS based Reply: P1L8: "Early" has been changed by early; P1L8: Yes, an EWS (early warning system) does not really avoid a disaster minimize disaster losses, so we change the description by "minimize disaster losses"; P1L9: in this paper we mean a type of landslides, so it should "be specific landslides"; P1L11 has been rephrased.

3. There are obviously substantial language issues that need to be resolved. Please revise. No further comments on typops and language are provided from my side in the remaining review. Reply: I have revised the language seriously.

4.P1L11: there is no explanation what a KF-FTT-SVM model is. What does the abbrevation stand for? P1L13: rather use landslide instead of slide. Reply: The full name of KF-FTT-SVM is given in P1L11. Slide is replaced by Landslide in P1L13.

5.Introduction: P1L21: Provide a citation for the claim that landslides are the third largest (?) geological hazard. P1L22: Add a space before the bracket. There are also several instances of missing spaces in the remaining document. P1L26ff: additional information on landslide EWS could be added, consider adding information from Overview on landslide EWS and review of existing systems: Thiebes, Benni, and Thomas Glade. "Landslide Early Warning Systems – Fundamental Concepts and Innovative Applications." In Landslides and Engineered Slopes. Experience, Theory and Practice, edited by S Aversa, L Cascini, L Picarelli, and C Scavia, 1903–1911. Naples, Italy: CRC Press, 2016. https://doi.org/10.1201/b21520-238. Thiebes, Benni. Landslide Analysis and Early Warning Systems: Local and Regional Case Study in the Swabian Alb, Germany. Springer Theses Series. Springer, 2012. Case studies: Thiebes, Benni, Rainer Bell, Thomas Glade, Stefan Jäger, Julia Mayer, Malcolm Anderson, and Liz Holcombe. "Integration of a Limit-Equilibrium Model into a Landslide Early Warning System." Landslides 11, no. 5 (June 14, 2013): 859–75. https://doi.org/10.1007/s10346-013-0416-2. Calvello, Michele, Ricardo Neiva d'Orsi, Luca Piciullo, Nelson Paes, Marcelo Magalhaes, and Willy Alvarenga Lacerda. "The Rio de Janeiro Early Warning System for Rainfall-Induced Landslides: Analysis of Performance for the Years 2010–2013." International Journal of Disaster Risk Reduction, October 2014. https://doi.org/10.1016/j.ijdrr.2014.10.005. Michoud, C., S. Bazin, L. H. Blikra, M.-H. Derron, and M. Jaboyedoff. "Overview of Existing Landslide Early-Warning Systems in Operation." In EGU General Assembly Conference Abstracts, 14:2919, 2012. http://adsabs.harvard.edu/abs/2012EGUGA..14.2919M. Piciullo, Luca, Michele Calvello, and JoséÂaMauricio Cepeda. "Territorial Early Warning Systems for Rainfall-Induced Landslides." Earth-Science Reviews 179 (April 2018):228–47. https://doi.org/10.1016/j.earscirev.2018.02.013. Piciullo, Luca, Mads-Peter Dahl, Graziella Devoli, Hervé Colleuille, and Michele Calvello. "Performance Evaluation of the National Norwegian Early Warning System for Weather Induced Landslides." Natural Hazards and Earth System Sciences Discussions, January 16, 2017, 1–28. https://doi.org/10.5194/nhess-2017-24. Rossi, Mauro, Silvia Peruccacci, M. T.Brunetti, I Marchesini, S. Luciani, Francesca Ardizzone, V Balducci, et al. "SANF: National Warning System for Rainfall-Induced Landslides in Italy." In Proceedings of the 11th International & 2nd North American Symposium on Landslides, edited by E Eberhardt, Corey R. Froese, A. Keith Turner, and S. Leroueil, 2:1895–99. London: Taylor & Francis, 2012. Segoni, S., A. Battistini, G. Rossi, A. Rosi, D. Lagomarsino, F. Catani, S. Moretti, and N. Casagli. "Technical Note: An Operational Landslide Early Warning System at Regional Scale Based on Space–Time-Variable Rainfall Thresholds." Natural Hazards and Earth System Science 15, no. 4 (April 16, 2015): 853–61. https://doi.org/10.5194/nhess-15-853-2015. Kirschbaum, Dalia Bach, Robert Adler, Yang Hong, Sujay Kumar, Christa Peters-Lidard, and Arthur Lernerlam. "Advances in Landslide Nowcasting: Evaluation of a Global and Regional Modeling Approach." Environmental Earth Sciences 66, no. 6 (July 2012): 1683–

96.https://doi.org/10.1007/s12665-011-0990-3. Reply: P1L21 it was said in a Chinese book, and I cannot found the citation, so I change the description. P1L22: Space is added. P1L26: I have down load all the papers and read them carefully. I have rewrote the introduction, and refresh the citation.

6. P2L10: you presented the UNISDR/UNDRR classification of EWS; however, the systems you mention in this line and the following are not necessarily real EWS that include works on all 4 fields of action. They are mostly monitoring and/or forecasting systems. Please check carefully whether they are really fully fledged EWS, e.g. by checking to which extent they are really in operation (most are not but are monitoring systems with some aspects of predictions). P3 Figure 1: Typos in figure. Seneor should be sensor. P5L8: can you provide more information on the characteristics of the landslide, the dam and the lake? What are the geological and geomorphological conditions? Why was the landslide triggered? Have there been any movements before? P11L5ff: the language is not appropriate. Please rephrase this section. Reply: P2L10: In this study we focus on the monitoring and warning model of landslide EWSs. I have emphasized in P2L5. P3: Figure has been changed. P5L8: More information is given in P5L10. P11L5: The language is rephrased.

7. Discussion: The discussion does not really reflect on the limitations and uncertainties of the study but rather summarises the research. This should be revised. Reply: The limitation and uncertainties are given in P16L28.

Please also note the supplement to this comment:
https://www.nat-hazards-earth-syst-sci-discuss.net/nhess-2019-48/nhess-2019-48-AC2-supplement.pdf

**Supplement:**

**Fast monitoring and real time predication method for early warning system in Baige landslide, Tibet, China**

Yongbo Wu, Ruiqing Niu*, Zhen Lu

Institute of Geophysics and Geomatics, China University of Geosciences, Wuhan, China

*Correspondence to: Ruiqing Niu,*E-mail address*: rqniu@163.com

**Abstract:** Landslide early warning system (EWS) have been widely used to minimize disaster losses. In this paper, a fast monitoring and real time predication method is proposed to build the Landslide EWS for specific landslides. The fast monitoring network in this system uses ad-hoc technology to build rapid on-site monitoring network consist of Beidou terminals, which is a technology similar to GPS, and fracture monitors. The real time predication method based on the combination of kalman filtering(KF), fast fourier transform(FFT) and support vector machine(SVM),short for KF-FFT-SVM, is conducted to make a real time precursor early warning short time before the occurrence of the landslide. The KF-FFT-SVM model working here is established by analysing the precursor landslide character in deformation data got by the Beidou terminals. The deformation data can be considered as the mechanical vibration of specific landslides, and the KF-FFT-SVM model is trained to predicate the occurrence of landslide by the deformation data. The fast monitoring technique improves the robustness of on-site monitoring system, and the real time predication method provides an effective early warning method for specific landslides. It is applied in Baige landslide, Tibet, China, monitoring and results show that the KF-FFT-SVM model can predication the occurrence of landslide with high accuracy. It will make the early warning work for specific landslides more effective if numerous continuous monitored precursor landslide deformation data are used to train the model well.

**1 Introduction**

Landslide hazard is the most common geological hazard in natural world. It is also direct affected by human engineering activities. China is one of the countries that suffered most from landslide disasters in the world(Huang, 2007). Especially after the Ms 8.0 Wenchuan earthquake in May 12, 2008, tens of thousands of landslides over a broad area in west China were triggered, some of which buried large sections of the towns and dammed the rivers(Dai et al., 2011). So the researches on reducing property damages and casualties have always been an urgent problem, and the EWSs which have already been working in many place of the world are an effective way for landslide early warning(Bach et al., 2012; Glade and Nadim, 2014; Piciullo et al., 2017, 2018; Stähli et al., 2015).

According to the definition of the United Nations International Strategy for Disaster Reduction (UNISDR 2009), an EWS is defined as "the set of capacities needed to generate and disseminate timely and meaningful warning information to enable individuals, communities and organizations threatened by a hazard to prepare and to act appropriately and in sufficient time to

reduce the possibility of harm or loss." Refer to the above definitions, efficient landslide EWSs should comprise four main sets of actions(DiBiago and Kjekstad, 2007): (1)Monitoring activities, i.e. data acquisition, transmission and maintenance of the instruments;(2)Analysis and modelling of the phenomenon;(3)Warning, i.e. the dissemination of simple and understandable information to the exposed elements;(4)Effective response of the elements exposed to risk and risk's

5    knowledge. In this study we focus on the monitoring and warning model of landslide EWSs.

Landslide EWSs can be divided into regional landslide EWSs and single landslide EWSs from scale range. In the regional landslide EWSs, the warning threshold is determined by statistic method. These systems are applicable for the rainfall induced shallow landslides, and the classification early warning is given according the preset rainfall intensity–duration threshold, also considering the soil moisture(Baum and Godt, 2010; Calvello et al., 2015; Gariano et al., 2015, 2016; Hong and Adler, 2007;

10   Rosi et al., 2015).

For single landslide EWSs, a successful EWS has the ability to measure and identify the significant indicators, called precursors, which precede a landslide catastrophic failure(Barla and Antolini, 2016). The precursor characters could be discovered from the mechanical properties of the landslide which are measured by instruments. For example, inclinometer for tilt(Dikshit et al., 2018; Lollino et al., 2002), fiber Bragg grating for fissures(Zhu et al., 2017), Ground-Based Synthetic-

15   Aperture Radar, LiDAR, total station, GPS and photogrammetric techniques for deformations(Atzeni et al., 2015; Barla and Antolini, 2016; Jaboyedoff et al., 2012; Malet et al., 2002; Tarchi et al., 2003), geoelectrical monitor for soil moisture(Supper et al., 2014), wire extensometer for rock fracture(Intrieri et al., 2012), etc. These precursor characters are used to make early warning with respective model or integrated models(Thiebes et al., 2014; Yin et al., 2010).

It is obviously that the warning model should be built according to the mechanism of the instability of a landslide. And

20   the predication accuracy of an early warning model relies on the high quality real-time monitoring data. In practice, the implement of on-site monitoring network always takes a long time as the design of the monitoring system is complex. Meanwhile, the on-site monitoring network is easy to broken down in the wild, which means the monitoring part of the most existing landslide EWSs are less robustness.(Intrieri et al., 2013).

In this paper, fast monitoring and deformation precursor predication method is proposed to improve the Landslide EWS.

25   The ad-hoc network technology is used to ensure the robustness of the monitoring part the early warning system. In order to build an on-site monitoring network quickly, especially after the first failure of a landslide, only Beidou terminal and fracture monitor are used to build the monitoring stations. The early warning part is based on KM-FFT-SVM model to make a real time predication. The fast monitoring system was applied after the Baige landslide first damming event, Tibet, and, successfully, got the critical slip data of the surface moving by Beidou terminal based on China's Beidou Navigation System. Then we use

30   the critical slip data to train the KM-FFT-SVM model and the following damming event is predicted successfully with the trained model. Practice shows the fast monitoring and real time predication method are of general significance.

**2 Fast monitoring system**

**2.1 Traditional monitor system**

The structure of a traditional landslide monitoring network is shown in Figure 1. All kinds of monitoring sensors are connected with data acquisition terminal (DTU) through Modbus protocol or SDI-12 protocol. DTU, communication module (GPRS/3G/4G) and power supply system constitute a remote measurement unit (RTU). The measuring data are sent to the mobile communication network through the communication module and transmitted to the control center through the public network. In this way, the system robustness is poor, because if the communication module of one monitoring point breaks down, all sensor data under this monitoring point will not be collected, which means partial paralysis of the monitor system. Therefore, a more flexible and stable networking structure is needed to improve the robustness of the monitoring system.

[Figure]

**Figure 1: Traditional landslide monitoring system**

**2.2 Ad-hoc network monitoring system**

The adaptive landslide monitoring network is based on ad-hoc network. Ad-hoc network is more secure, robust, stable and reliable comparing with traditional bus and star network by using adaptive technology. Figure 2 is the typical structure of adaptive landslide monitoring network. In here, four stations are listed and it can be expanded if more stations are needed in practical application. Each station is composed of several sensors, a data acquisition instrument and an ad-hoc router. Each ad-hoc router has the communication module of GPRS/3G/4G. At the same time, each ad-hoc router forms a local ad-hoc network through LoRa technology. The ad-hoc router acts as an AP (Access Point) node, which is responsible for the access of the external network. At the same time, the ad-hoc routers can also communicate with each other by multi-hops. When one node

breaks down, the network will find other paths to bypass this node through routing algorithm, which improves the network's robustness. The adaptive landslide monitoring network has three working modes. One is Normal mode, as is shown in figure 3(a); Another is Communication fault mode, as is shown in figure 3(b). In this mode, the GPRS/3G/4G communication modules in some of the ad-hoc routers cannot work, so the system finds the new routing path to send the data out; The third is

5  Beidou satellite communication mode, as is shown in figure 3(c). This mode means the GPRS/3G/4G communication modules in all of the ad-hoc routers cannot work, so the beidou satellite communication system will be started. With the advantage mentioned above, the ad-hoc network landslide monitoring system could be built immediately with high robustness, especially in the place where there are no mobile signals or the signals is weak.

[Figure]

10  **Figure 2: Ad-hoc network monitoring system**

[Figure]

**Figure 3: Adaptive landslide monitoring network working mode. Normal mode (a), Communication fault mode (b), Beidou satellite communication mode (c).**

**2.3 Application of the fast monitoring system**

5    In the early morning of October 11, 2018, a large-scale high-level landslide occurred on the Tibetan Bank of Jinsha River at the junction of Baige Village, Boro Township, Jiangda County, Tibet Autonomous Region, and Zeba Village, Ronggai Township, Baiyu County, Sichuan Province, which blocked the main stream of Jinsha River and formed a barrier lake. Then, on the late day of November 3, second landslide occurred and blocked the Jinsha River again. The barrier lake formed by the twice landslides is a great threaten to the people live in the lower reaches of Jinsha River. The location of the landslide is

10    shown in figure 4.The mountain body near the Baige landslide is made of metamorphic rock. The rock in the upper part is soft, while the lower part is hard. The weak fractured rock mass affected by tectonics deforms under the action of long-term gravity and eventually loses its stability. The historical remote sensing image shows that the rocks in Baige landslide area has undergone deformation for at least 50 years, and the surface displacement in some areas is close to 50 m. There are approximate $2.2 \times 10^7 m^3$ rock and soil fall into the Jinsha River  and form a dam river in the first landslide.  The length of the dam along

15    the valley is about 1100 m, and the width of the dam is about 500 m in the vertical direction. The maximum height of the dam over the original river surface is about 85 m, and the average thickness is 40 m. The second landslide occurred at the back edge of the first landslide. The total volume of the rock fell down is about $8.5 \times 10^6$ m$^3$. The unstable rock mass scrapes the broken rock mass along the way, forming debris flow, and blocking up the Jinsha River again. The weir dam is 50m higher than the one formed by the first landslide, the volume of accumulation rock is $9.3 \times 10^6 m^3$(Qiang et al., 2018).

[Figure]

**Figure 4: Location of  Baige landslide.**

Baige landslide occurred suddenly, and there are no monitor device working there before. While the monitoring system need to be built immediately to ensure the safety of the emergency rescue works for dredging Barrier Lake. As there are no
5   mobile signal there, the fast monitoring system is applied there. We use surface displacement monitoring equipment and

fracture monitoring equipment to build the fast monitor system, as shown in Figure 5. Figure 5(a) show the Beidou receiver which is the surface displacement monitoring equipment. Figure 5(b) show the fracture monitor which is the fracture monitoring equipment. Both of them use solar panels as an option for energy supply. The locations of these monitor equipment are showed in figure 6. BD1, BD2, BD3 and BD4 represent Beidou receiver, while FM1, FM2, FM3, and FM4 represent fracture monitor. The sensors are located nearby the back edge fault scarp to make sure to get the real deformation data of the landslide.

(a) Beidou receiver       (b) Fracture monitor

[Figure]

**Figure 5: Monitor device on the landslide.**

[Figure]

**Figure 6: Locations of the equipment on the landslide**

**3 KF-FFT-SVM model**

**3.1 Kalman filtering**

[revised manuscript text omitted]
 researches predicate the deformation of landslide based on these factors. In this paper, we don't have too many prior knowledge about the Baige landslide. The only quantitative data we got is are deformations measured in finite time. So in the KF-FFT-SVM landslide early warning model, firstly, we used the finite deformation data sequence $S_k$ got by Beidou receivers to build the input of kalman filtering predication model $X_k = (S_k, V_k, A_k)$, where $V_k$ and $A_k$ are the velocity and acceleration of $S_k$, respectively. After the first step we got the predication and filtering result of $A_n$. Formula (21) is the precision evaluation of kalman filtering predication model. Secondly, we use FFT to analysis the spectrum characteristics of the deformation acceleration sequence $A_n$ nearby the occurring time of landslide, and find the precursor character of the Baige landslide 'Step $k$' which represents the precursor frequency character of $A_n$. Finally, we use $A_n$ form the training data and testing data according to the precursor character 'Step $k$'and label them. Then the SVM model is trained with the training data and the trained SVM model will be used to make predication by a new $A_k$ to find out if the warning is made at that time. The predication result $B_n$ is a vector with the same dimension of $A_k$ and its value is either '0'or '1'. '0' represents that there is no warning while '1' represent that there is a warning. The precision of classification result is given as formula (22). The whole process of KF-FFT-SVM landslide early warning model is showed in figure 7.

$$\text{RMS} = \frac{1}{M}\sqrt{\sum_{i=1}^{N}(X_i - Z_i)^2} \tag{21}$$

$$Accuracy = \frac{Right\ classification\ numbers}{whole\ samples} \tag{22}$$

[Figure]

**Figure 7: Process of KF-FFT-SVM model**

**4 Application of real time predication method**

5  **4.1 Data pre-processing**

The deformation data of Baige landslide after its first slide were gotten by Вeidou receivers. The second landslide occurred on late November 3, so we choose the date from October 31 to November 6 to train the KF-FFT-SVM early warning model. Figure 8 shows the raw data of BD1, BD2, BD3 and BD4 and the horizontal deformation data are used here. The raw deformation data are gotten at the interval of 10 minutes. It is know from fig.8 that there are noise in the data and

10  the deformation data gotten on much points at 10 minutes interval are unchanged. So we make a 30 minutes statistics, which means a 30 minutes interval sampling. The 30 minutes statistics result is shown in figure 9. The statistical data are used to build the kalman filttering model.

[Figure]

**Figure 8: Raw data of deformation.**

[Figure]

**Figure 9: 30 minutes statistics of the raw deformation data**

**4.2 KF-FFT-SVM model analysis**

**4.2.1 KF model build**

In built landslide deformation KF model, we choose deformation, deformation velocity and deformation acceleration as the state vectors, which are $S_k$, $V_k$ and $A_k$ respectively .The relations between them are:

$$\begin{cases} X_k = [S_k, V_k, A_k]^T, \\ S_k = S_{k-1} + V_{k-1} \cdot T_s + w_{k-1}^1, \\ V_k = V_{k-1} + A_{k-1} \cdot T_s + w_{k-1}^2, \\ A_k = A_{k-1} + w_{k-1}^3. \end{cases} \tag{23}$$

Where $T_s$ is the data acquisition interval; $w_k^1, w_k^2, w_k^3$ are random errors. Let $T_s = 1$, then the stochastic difference equation of system state is:

$$X_k = \begin{bmatrix} 1 & 1 & 0 \\ 0 & 1 & 0 \\ 0 & 0 & 1 \end{bmatrix} X_{k-1} + [w_{k-1}^1, w_{k-1}^2, w_{k-1}^3]^T \tag{24}$$

The observation formula is described as formula (25):

$$Z_k = [1\ 1\ 0]X_{k-1} + v_{k-1} \tag{25}$$

Where $v_k$ is random error, and we got $A = \begin{bmatrix} 1 & 1 & 0 \\ 0 & 1 & 0 \\ 0 & 0 & 1 \end{bmatrix}$, H= [1,1, 0]. The random error $w_k$ and $v_k$ are unknown, and our purpose is to use formula (5) ~ (11) to determine the solution of them. At the beginning, we can set a random value of Q and R, then use the date in section 4.1 to find a couple value of Q and R that makes the KF model converge to optimal solution. Figure 10 is the fitting result of BD1, BD2, BD3 and BD4, seeing the red curve. In this KF model $W_k=[5,3,3]^T$, $V_k=3$. The max fitting error is 5.73mm which means the built KF model has a good prediction and filtering result.

[Figure]

**Figure 10: Kalman filtering fitting result**

**4.2.2 FFT analysis**

In the FFT analysis, we choose 64 deformation acceleration data gotten by section 4.2.1 between November 3 and November
5    4, which is the time close to the moment the secondary landslide happening, to conduct FFT analysis. Figure 11 show the FFT
result of deformation acceleration date during the precursor stage. The FFT length N is 64 and the acquisition interval is $T_s$.
Let $T_s = 1$, the acquisition frequency can be simplified to 1 Hz. In figure 11, there are two major amplitude peak value nearby
0.2 Hz and 0.9 Hz, which means, in time domain, the precursor landslide character period is nearby $5T_s$. So we choose the step
sequence $k = 2, 3, 4, 5, 6, 7, 8$ to construct deformation acceleration sequence.

[Figure]

**Figure 11: FFT of precursor landslide character**

**4.2.3 SVM model training**

Before SVM model is trained, we mark the deformation acceleration data of BD1, BD2, BD3 and BD4 manually. The data between November 3 and November 4 are marked with label "+1" which represents the precursor landslide character, others are marked with label "-1" which represents the non-precursor landslide character. Then, we use the marketed BD1, BD2, BD3 and BD4 data respectively to construct acceleration sequence $A'_n$,

$$A'_n = [A_n, A_{n+1}, \dots, A_{n+k-1}, label] \tag{26}$$

where $k = 2,3,4,5,6,7,8$, which is given in part 4.2.2; $n = 1,2,\dots,336-k$; label is marked value of $A_n$.

Then we get the marked set $A'_{n(BD1)}$, $A'_{n(BD2)}$, $A'_{n(BD3)}$, and $A'_{n(BD4)}$. Choose $A'_{n(BD1)}$, $A'_{n(BD2)}$ and $A'_{n(BD3)}$ as training data, while $A'_{n(BD4)}$ as testing data. For example, when $= 2$, the training data and testing data is showed in formula (27):

$$\begin{cases} & A'_{n(BD1)} = [A_n, A_{n+1}, label]_{(BD1)} \\ training\ data: & A'_{n(BD2)} = [A_n, A_{n+1}, label]_{(BD2)} \\ & A'_{n(BD3)} = [A_n, A_{n+1}, label]_{(BD3)} \\ testing\ \ data: & A'_{n(BD4)} = [A_n, A_{n+1}, label]_{(BD4)} \end{cases} \tag{27}$$

**4.3 Predicting result**

Use RBF function and SMO algorithm to search the best *C* and γ.The predicting results at different step sequence are showed in figure 12. Figure 12 also show the 3 steps sequence training data scatters in 3D coordinate and it is obvious to know that they cannot be well separated in 3D space. So we should separate them in high dimension, which means k>3. From figure 12, we can know that when k=6, the optimal result is gotten and the highest accuracy =0.915, the best *C*=4, and γ = 1. The result is proved by the FFT analysis that the best precursor landslide character is nearby 0.2Hz, which equal to 6 steps sequence in SVM model.

**5 Discussion**

The fast monitoring and real time predication method in this paper focus on the monitoring, warning model of the landslide EWSs. Effective response manners is beyond the discussion of this paper. The use of ad-hoc technology help to build a redundancy on-site monitoring network, which improves the robustness of the landslide EWSs. The monitoring system built here only includes Beidou terminals and fracture monitors. So it can be built in a short time, and it will be useful for the landslide monitoring immediately after the first-time failure of a landslide, because the secondary landslide may occur at any time, which is a great threat to rescuers.

The real time predication method based on the KF-FFT-SVM model makes the predication according to the acceleration characters of landslide deformation. It is on the principle that the mechanical vibration of landslide failure can be recorded by the deformation data. Then the precursor deformation acceleration characters is considered as the vibration of landslide failure. In this study, the Beidou terminals have the ability to record the deformation of landslide in a short time (10 minutes) with an accuracy of a few millimetres which is given by the manufacture. The raw deformation data are showed in figure 8, it is obviously that there are random errors gotten by the Beidou terminals as the deformation data are not continuous rise in several periods. That's why the raw data are pre-processed and the 30 minutes statistics of the raw deformation data are, then, given in figure 9. Suppose 30 minutes as one unit time scale, then KF method is used to filter the random error, and FFT is used to find the precursor deformation acceleration characters which represent the mechanical vibration frequency of landslide failure. Finally, the characteristic frequencies gotten by the FFT method are different between BD1, BD2 and BD3, then several frequencies are used to generate different length time sequences. SVM model is trained by these sequences respectively and the accuracy is tested by BD4 to find the best character frequency which can be used to make the early warning for this specific landslide.

There are limitations and uncertainties in the application of real time predication method. The most important features of the method is that it can quickly build monitoring network and uses the deformation data to carry out precursor landslide early warning. The problem is that, in KF-FFT-SVM model, we consider the deformation data measured by Beidou terminals as the mechanical vibration of landslide. While, there must be distortion for the transformation from vibration signal to deformation signal. Meanwhile, the monitoring position is also a key factor for the authentic record of the precursor landslide characters.

In practice it is difficult to put the measuring instrument directly on the top of deformation place. We can only locate the Beidou terminals near the surface fracture to make sure that the precursor landslide characters are recorded as authentic as possible. In this study, we only use finite data to train the model, and the method will be more effective if there are more data sets used to train the warning model.

In the future study, we will do researches on the relationship between the surface deformations and inner vibration of the landslide. By doing so, rock and soil press sensor will be used to measure the authentic vibration of the landslide. We will also build a data base of the landslide precursor monitoring data and use landslide monitoring data by other type of sensors to train the KF-FFT-SVM model.

**6 Conclusion**

In this study, the fast monitoring and real time predication method for landslide is proposed. This method uses ad-hoc technology to facilitate the repaid building of the on-site monitoring network, which improves the robustness of traditional landslide EWSs. Furthermore KF-FFT-SVM model is built to make real time early warning for single landslide through the analysis of the precursor landslide characters from the deformation data. The KF-FFT-SVM model trained by the precursor deformation data of landslide makes the single landslide early warning more effective and it also can be combined with the monitoring of the acoustic emissions from a specific landslide(Hu et al., 2018).

The most important features of the method is that it can quickly build a monitoring network and uses the deformation data to make precursor landslide early warning. This is very useful for the early warning of the specific landslide after its first-time failure. It provides a new idea for monitoring and early warning of single landslide, which not only improves the robustness of the landslide early warning system but also makes landside early warning does not depend too much on the study of landslide mechanism characteristics and the monitoring of many landslide elements. Another innovation of the study is that we extracte precursor characters, which are considered as the mechanical vibrations of the landslide failure, from the deformation data, then make the real time early warning according to the precursor deformation data of landslide.

[revised manuscript text omitted]

Thiebes, B., Bell, R., Glade, T., Jäger, S., Mayer, J., Anderson, M. and Holcombe, L.: Integration of a limit-equilibrium

5    model into a landslide early warning system, Landslides, 11(5), 859–875, doi:10.1007/s10346-013-0416-2, 2014.

Yin, Y., Wang, H., Gao, Y. and Li, X.: Real-time monitoring and early warning of landslides at relocated Wushan Town, the Three Gorges Reservoir, China, Landslides, 7(3), 339–349, doi:10.1007/s10346-010-0220-1, 2010.

Zhu, H. H., Shi, B. and Zhang, C. C.: FBG-based monitoring of geohazards: Current status and trends, Sensors (Switzerland), 17(3), doi:10.3390/s17030452, 2017.

10   Sättele, M., Bründl, M., and Straub, D.: A classification ofwarning system for natural hazards, in: 10th International Probabilistic Workshop, edited by: Moormann, C., Huber, M., and Proske, D., Stuttgart: Institut für Geotechnik der Universität Stuttgart, 257– 270, 2012.